# Effects of a Feed Sanitizer in Sow Diets on Sow and Piglet Performance

**DOI:** 10.3390/ani15243618

**Published:** 2025-12-16

**Authors:** Sara Williams, Francisco Domingues, Hayford Manu, Andres Gomez, Lee Johnston

**Affiliations:** 1Department of Animal Science, University of Minnesota, St. Paul, MN 55108, USA; swilli54@uoguelph.ca (S.W.); gomeza@umn.edu (A.G.); 2Anitox Corporation: North America (Headquarters), Lawrenceville, GA 30043, USA; fdomingues@anitox.com; 3Southern Research and Outreach Center, University of Minnesota, Waseca, MN 56093, USA; manu0063@umn.edu; 4West Central Research and Outreach Center, University of Minnesota, Morris, MN 56267, USA

**Keywords:** feed sanitizer, formaldehyde, sow, piglet, gut microbiome

## Abstract

Feed contaminated with harmful bacteria and other microorganisms can negatively affect pig health and production. This study investigated whether adding a feed sanitizer containing formaldehyde and propionic acid to pregnant and nursing sow diets would improve performance and health outcomes for both sows and their offspring. One hundred and seven sows were divided into two groups: one group received regular feed while the other received feed with the sanitizer added. The sanitized feed was provided from late pregnancy through nursing until the piglets were weaned at about 19 days old. Researchers measured sow body condition, feed consumption, breeding efficiency, litter performance, and piglet health and analyzed gut bacteria communities in both sows and piglets. Results showed that the feed sanitizer did not significantly improve sow performance, piglet growth, or piglet health, though it did reduce the weight of stillborn piglets. The sanitizer caused some changes in gut bacteria populations, but these changes did not translate into measurable performance benefits. While feed sanitizers may have theoretical benefits for reducing harmful microorganisms in pig feed, this study did not demonstrate clear advantages under the conditions tested, suggesting that more research is needed to understand when and how feed sanitizers might be most beneficial.

## 1. Introduction

Feed sanitizers are a type of feed additive that, when applied to or mixed into feed, provide an antimicrobial benefit. Feed sanitization addresses significant economic and health challenges in swine production. Animal feed serves as a reservoir for various pathogens that can enter feed through multiple pathways including ingredients, dust, contaminated equipment, and wildlife [1,2,3,4]. Feed sanitization also prevents mold growth and mycotoxin development. Molds reduce feed quality and can produce mycotoxins that pose serious health risks to animals [5]. Assorted feed sanitizing practices can be used to prevent or remove mycotoxins in ingredients for animal feed [6]. Several authors have demonstrated that the inclusion of formaldehyde-based sanitizers in feed reduces the concentration of pathogens [7,8] and molds [8] present in feed. The reduction in feed-borne pathogens reduces the exposure of animals to these pathogens which reduces disease pressure. Similarly, the reduction in feed-borne molds decreases the possibility of mycotoxin contamination of feeds consumed by livestock. The reduction in pathogens and mycotoxins in feed supports the good health of livestock.

The impact of feed sanitizers, specifically formaldehyde-based sanitizers, on animal performance and health parameters requires careful examination. Formaldehyde occurs naturally throughout the environment and in many common foods. As an intermediate product in biological metabolism, ingested formaldehyde is metabolized rapidly in the bloodstream into formic acid, which is further broken down into carbon dioxide and water [9]. With a metabolic half-life of only 60–90 s [4,10], formaldehyde is quickly eliminated from the system of the animal. This efficient metabolic pathway provides reasonable assurance of safety for animals when used appropriately.

Research on formaldehyde-based feed sanitizers shows variable effects on pig performance depending on the application method and inclusion level [7,11,12]. Beyond performance impacts, formaldehyde treatment can significantly alter the gut microbiome composition of nursery pigs. Williams et al. [13] reported that formaldehyde in feed decreased beneficial taxa such as Lactobacillaceae and Paraprevotellaceae, and increased abundances of potentially harmful taxa from the Clostridaceae and Erysipelotrichaceae families in the gut microbiome. This altered microbial community structure suggests that while formaldehyde treatment effectively reduces the bacterial contamination of feed, it may create unintended consequences for the gut microbiome that could detrimentally impact pig health and performance [13].

The documented effects of formaldehyde-based feed sanitizers on nursery pig performance and microbiome raise important questions about their safety profile in breeding animals, especially sows. However, there are no reports of feeding formaldehyde-based sanitizers to breeding swine. Building on both the established safety profile and the observed variable effects on the performance of pigs and the gut microbiome, our study aims to fill a critical knowledge gap by examining how a formaldehyde-based feed sanitizer specifically affects sows and their litters. We hypothesize that including a formaldehyde-based feed sanitizer in sow diets will improve both maternal performance and nursing litter outcomes while inducing distinct changes in the sow and piglet microbiome profiles.

## 2. Materials and Methods

The protocol for this experiment was reviewed and approved by the University of Minnesota’s Institutional Animal Care and Use Committee (IACUC# 2305-41095A). The experiment began in August 2023 and concluded in November 2023. This experiment was conducted at the University of Minnesota’s Southern Research and Outreach Center (SROC) in Waseca, MN, USA.

### 2.1. Animals, Housing, and Treatments

Two consecutive farrowing groups of mixed-parity sows (TN70 females; parity range = 0 to 8; Topigs Norsvin, Bloomington, MN, USA) were divided based on breeding date into group one (*n* = 53) and group two (*n* = 54). Within each group, sows were assigned to one of two dietary treatments and began receiving their assigned diet at day 80 of gestation. Parity of sows at assignment was equalized across groups and dietary treatments as much as possible. Sows were selected based on parity to prevent confounding effects of parity on performance response criteria and sample analyses.

Dietary treatments consisted of: (1) Control—sows fed a commercially relevant corn-soybean meal-based diet during gestation and lactation and, (2) Treatment—Control + 0.55% of a feed sanitizer based on a blend of formaldehyde and propionic acid (Table 1). Dietary treatment was imposed by adding 5.5 kg of the feed sanitizer powder per metric ton to complete feed at the expense of corn. All sows received 2.72 kg of feed once daily during gestation and remained on their assigned dietary treatments through parturition. After parturition, sows were allowed ad libitum access to their assigned lactation diet and water.

Beginning at about day 42 of gestation, sows were housed in group gestation pens (about 55 sows/pen) on totally slatted concrete floors. Sows received their feed daily via an electronic sow feeder (TEAM system; Osborne Industries, Osborne, KS, USA). At approximately day 109 of gestation, sows were washed and moved to individual farrowing stalls (2.13 m × 0.97 m × 0.66 m) within farrowing rooms until weaning of litters. Each farrowing room contained 16 stalls. Each farrowing group was housed across four rooms, with a total of eight rooms being used in the study. An equal number of Control and Treatment sows were housed in each farrowing room. Farrowing stalls were equipped with one dry sow feeder, one nipple waterer, and a slatted cast iron floor over a deep manure collection pit. The remaining side areas of the stall featured a plastic-coated floor, a heat lamp, and a water nipple drinker for piglets. Heaters and ventilation fans were operated via a computerized controller (Edge System, EDGE 3-slot expansion, GSI Group LLC, Armand Frappier Saint-Hubert, QC, Canada). The temperature in the farrowing room was maintained at 20 ± 1 °C. During farrowing, additional heat was provided to piglets via heat lamps and floor heating pads.

The standard operating procedures for piglets included tail docking, administration of an intramuscular injection of iron (1 cc, 200 Fe/cc VetOne**^®^**, Boise, ID, USA), and cutting and disinfecting the umbilical cord within 24 h post-farrowing. Between 5 and 8 days post-farrowing, male piglets were surgically castrated. All piglets were vaccinated against porcine circovirus disease 4 days prior to weaning (1 cc, Ingelvac CircoFlex, Boehringer Ingelheim, Ridgefield, CT, USA). Creep feed was not offered to piglets; however, piglets did have access to the sow’s feed.

Weaning of piglets occurred between 08:00 h and 10:00 h when piglets were approximately 18 ± 4 d of age. After all piglets were weaned, sows were transferred to the breeding/gestation barn, where they underwent estrus detection using a mature boar. Estrus was detected once sows displayed signs of standing heat and immobilization after boar exposure. The weaning-to-estrus interval was calculated and recorded. Estrus detection continued for up to 34 days post-weaning.

### 2.2. Sow and Piglet Performance

#### 2.2.1. Sows

Sows were identified individually using ear tags. At about days 80 and 109 of gestation, 24 h post-farrowing, and at weaning, sows were weighed. Sow backfat thickness was measured at the last rib via ultrasound (Renco Leanmeater, Minneapolis, MN, USA) on about day 109 of gestation, 24 h post-farrowing, and at weaning. Body condition of sows was assessed on day 109 of gestation and at weaning using the Knauer sow caliper [14] and recorded to the nearest half number. Feed disappearance was measured throughout lactation, and it was assumed that feed disappearance equaled feed intake as feed wastage was minimal. Feed consumption by piglets was considered negligible. Weekly feed intake for each sow was determined by adding individual daily feed intakes within sow.

Measures of sow reproductive performance included the following: total number of piglets born alive, number of stillborn piglets, number of mummified piglets, number of piglets after cross-fostering, and number of piglets at weaning per litter. Also, total litter birth weight before and after cross-fostering, the weight of mummies and stillborns, and the litter weight at weaning were collected. Pre-weaning piglet mortality was recorded.

#### 2.2.2. Piglets

Within 24 h of farrowing, litter sizes were standardized to approximately 14 piglets as much as possible. Piglets were cross-fostered within dietary treatments. Stillborn and mummified piglets were included in the count of total pigs born per sow. Pre-wean mortality was calculated by dividing the number of piglet deaths within the litter after cross-fostering by the total number of piglets within the litter after cross-fostering. Piglets were monitored daily for instances of morbidity and mortality. Any piglets that died during the study were weighed and the date and suspected cause of death were recorded. The cause of piglet deaths was categorized into six categories: nonviable, euthanasia, runt, injury, sickness, and scours. Causes of death were determined by barn staff. No necropsy examinations were performed. On the day of weaning, piglets were weighed, and the total litter weight was recorded.

Fecal samples from piglet scouring events were collected as they occurred and stored in a commercial freezer at SROC. Samples were then placed on dry ice and sent to the Veterinary Diagnostic Lab (VDL) at the University of Minnesota for enterotoxigenic *Escherichia coli* screening. Scouring events were determined based on a subjective fecal scoring method. Samples from a scouring pig or litter were scored on a scale of 0–3: 0 = solid; 1 = semi-solid; 2 = semi-liquid; and 3 = liquid. Piglets were deemed as no longer scouring when they produced feces with scores of 0 or 1 for at least 24 consecutive hours.

### 2.3. Feed Sample Analysis

Both Control and Treatment diets were manufactured at the on-site feed mill at SROC. The feed sanitizer powder was Termin-8**^®^** and was provided by Anitox Corporation (Lawrenceville, GA, USA). Termin-8 consisted of 18% formaldehyde and 4.7% propionic acid. Control sows did not receive any feed with added sanitizer. A feed sample was collected from each batch of feed and sent to Anitox Corporation to ensure the proper concentration of sanitizer in the feed. All samples were stored at −20 **°**C until shipment for analysis.

### 2.4. Microbiome Sample Collection

#### 2.4.1. Sows

Rectal swabs were collected from all sampled sows at about day 80 of gestation before dietary treatments were imposed, and at weaning. Sows to be sampled were selected randomly within parity and standardized across parities before collection. Briefly, a sterile cotton swab was used to scrape the rectal wall of each sow until the swab was impregnated with feces. The sample-laden swab was then placed in sterile collection tubes (5 mL; Thomas Scientific, Minneapolis, MN, USA). All samples were placed immediately on dry ice after collection until they could be frozen at −80 °C for subsequent DNA extraction.

For Control sows, 67 fecal microbiome samples were successfully collected and processed for DNA extraction, with 4 samples subsequently discarded prior to sequencing due to insufficient read depth. In total, 63 fecal microbiome samples from Control sows were used for final analysis. For Treatment sows, 61 fecal microbiome samples were successfully collected and subjected to DNA extraction. Of these, 3 samples were discarded prior to sequencing due to insufficient read depth, and an additional 3 samples were removed during bioinformatic processing. In total, 55 fecal microbiome samples from Treatment sows were used for final analysis. Ultimately, across both diets, 118 sow fecal microbiome samples were successfully extracted, sequenced, and utilized in the study.

#### 2.4.2. Piglets

At farrowing, two piglets per litter closest to the average birth weight of pigs in the litter were selected from 30 litters per treatment for future microbiome analysis. Fecal microbiome samples were not collected from piglets at farrowing. At selection, these piglets were ear notched and did not receive injectable antibiotics throughout lactation, unless necessary to ensure acceptable animal welfare. If a selected piglet was administered antibiotics, it was no longer considered for sample collection. At weaning, one ear notched piglet that was closest to the average weaning weight of piglets in the litter was sampled as described above for sows. Samples were collected from 24 Control litters and 21 Treatment litters, all of which underwent DNA extraction. For Control litters, 2 samples were subsequently excluded (1 prior to sequencing due to insufficient depth and 1 during bioinformatic processing). From Treatment litters, 4 samples were removed during bioinformatic processing. This resulted in 22 Control and 17 Treatment litter samples available for final analysis.

### 2.5. DNA Extraction and Sequencing

Microbial DNA was extracted from sow and piglet fecal samples using the ZymoBIOMICS DNA Miniprep Kit (Zymo Research Corp, Irvine, CA, USA) according to manufacturer protocols. For all samples, 16S rRNA sequencing was implemented. Negative controls, consisting of a sterile blank cotton swab, were created for each sample extraction period. The quality and quantity of extracted DNA were evaluated using a Nanodrop One Microvolume UV-Vis Spectrophotometer (Thermo Fisher Scientific, Waltham, MA, USA) before being sent for sequencing.

DNA extraction kits introduce their own microbiome into collected samples, creating potential contamination issues. For this experiment, negative controls (*n* = 17) consisting of a sterile swab, not exposed to the SROC environment, and each kit’s reagents were added during the DNA extraction for the identification and subsequent removal of potential reagent or environmental contaminants. Blank samples (*n* = 5) were chosen at random as sequencing controls.

Sequence data were generated through targeting the V4 variable region of the 16S rRNA bacterial gene on the MiSeq sequencing platform (MiSeq 2 × 300) using the primers F (5′-GTGYCAGCMGCCGCGGTAA-3′) and R (5′-GGACTACNVGGGTWTCTAAT-3′) and dual-indexing library preparation [15]. All sequence data were then processed using the Quantitive Insights Into Microbial Ecology 2 (QIIME2) pipeline [16] and its Divisive Amplicon Denoising Algorithm 2 (DADA2) plugin [17]. This procedure allowed us to process raw sequence data by removing primers, adaptors, and low quality sequences, assign taxonomy to amplicon sequence variants (ASVs) identified, and estimate their abundances in each sample using the Greengenes 13_8 reference database [18].

Predictive functional profiling of microbial communities was conducted using Phylogenetic Investigation of Communities by Reconstruction of Unobserved States (PICRUSt) analysis [19]. The resulting predictive pathway abundance data were exported to R Studio (version 4.5.1) for statistical analysis.

### 2.6. Statistical Analyses

#### 2.6.1. Sow and Piglet Performance

Performance data were analyzed using the GLIMMIX procedure of SAS (version 9.4, SAS Institute, Inc., Cary, NC, USA). Sow and litter were considered the experimental unit. For data collected repeatedly over time (body weight, backfat thickness, caliper measurements, feed intake, and litter characteristics), the statistical model included treatment, time, and their interaction as fixed effects. Parity and initial body weight were included in the model as covariates, where appropriate, and contemporary farrowing group was used as a random effect. For non-repeated data, the statistical model included treatment as a fixed effect, parity as a covariate, and contemporary farrowing group as a random effect. Categorical data (wean-to-estrus interval) were analyzed using the FREQ procedure with chi-square analysis. Significant treatment differences were indicated at *p* < 0.05, with trends recognized in the range of 0.05 ≤ *p* ≤ 0.10. All reported means are least squares means.

#### 2.6.2. Microbiome Data

All statistical analyses of the processed 16S rRNA sequence data were performed using the R statistical interface (www.r-project.org). The vegan package in R (Version 2.6-10) [20] was employed for alpha diversity analyses—Shannon Diversity Index, Chao1 richness, and Rarefied Richness—beta diversity analyses—Bray–Curtis distances—and Permutation Analysis of Variance (PERMANOVA) computations. The ape package [21] in R was utilized to conduct principal coordinate analyses (PCoA) based on the Bray–Curtis distances.

Analyses were conducted across three major comparisons: temporal changes within the Control diet (Day 80 gestation vs. weaning), temporal changes within the Treatment diet (Day 80 gestation vs. weaning), and diet effects at weaning (Control vs. Treatment).

To identify statistically significant and biologically relevant taxa and predicted metabolic pathways distinguishing treatment groups, Multivariate Association with Linear Models (MaAsLin) [22] was employed with stringent filtration criteria (*p* ≤ 0.05, *q* ≤ 0.005). The MaAsLin2 R procedure accounted for confounding variables. For temporal comparisons, the analysis incorporated collection time point as a fixed effect, while treatment comparisons at weaning utilized dietary treatment as a fixed effect. All analyses included farrowing group as a random effect. For visual representation and comparative analysis, the five most discriminant pathways for each dietary group at gestation and weaning were selected and presented in figures.

Visualization of microbial community composition was achieved through heatmaps generated using the pheatmap R package (Version 1.0.12) [23]. Alpha diversity and individual taxa comparisons were visualized using box plots created with base R functions. Statistical significance in alpha diversity and beta diversity was assessed using non-parametric tests including Wilcoxon tests for pairwise comparisons and PERMANOVA for community-level (multivariate) analyses, respectively. When parity alone provided insufficient power, parity was included as parity group. Parity group 1 was classified as parities 0–1, parity group 2 was parities 2–3, and parity group 3 was parities 4+. Results from MaAsLin2 were considered significant when the FDR-adjusted *p*-value (*q*-value) was <0.10. Statistical significance was denoted as follows: *** *p* < 0.001, ** *p* < 0.01, * *p* < 0.05, † 0.05 < *p* ≤ 0.10.

## 3. Results

### 3.1. Chemical Analysis

The average feed sanitizer concentration of Treatment gestation diets was 6.22 kg per metric ton of feed which contained 1118 ppm of formaldehyde and 292 ppm of propionic acid. Treatment lactation diets sampled at the time of delivery to the sow barn averaged 7.72 kg per metric ton of feed, across both groups one and two (Table 2). Average formaldehyde and propionic acid concentrations were 1390 and 363 ppm, respectively. Control diets had no sanitizer powder recovered.

### 3.2. Sow and Piglet Performance

Dietary supplementation with sanitizer during late gestation and lactation showed no significant effect on sow or piglet performance. No significant differences in average daily feed intake were observed between the Control and Treatment diets across individual weeks or the entire lactation period (Table 3). Compared to Control sows, body weights of sows fed the Treatment diet were significantly lower throughout the experimental period (*p* < 0.01). Backfat depth measurements revealed no significant differences between treatments at day 80 of gestation, day 109, within 24 h post-farrowing, or at weaning. Caliper measurements showed no significant differences between treatment groups throughout the experimental period. As expected, parity accounted for a significant portion of the variation in sow weight, backfat depth, and caliper measurements.

The average wean-to-estrus interval did not differ significantly between Control and Treatment sows (Table 4). A total of 98% of Control sows and 94% of Treatment sows displayed estrus by day 7 post-weaning. This distribution pattern remained consistent through day 14 and day 21 post-weaning. Chi-square analysis revealed no significant effect of feed sanitizer supplementation on the timing of estrus expression within the 21-day post-weaning period.

Analysis of litter performance metrics revealed no meaningful impact attributable to feed sanitizer (Table 5). No statistically significant differences were observed in litter size at birth or weaning between treatments. Compared to Control, the number of stillborn piglets per litter was lower in litters farrowed by sows fed the Treatment diet. The weight of mummies per litter was significantly higher in Control litters. Individual piglet weights at birth and weaning, as well as overall litter weight gain, were not significantly different between dietary treatments. Piglet mortality rate was not different between litters nursing from Control versus Treatment sows.

The incidence of scours observed throughout the study period was very low, as only three scouring events were recorded across both treatments. All scour samples were subjected to pathogenic *E. coli* analysis to elucidate potential scouring causes. All samples showed no presence of pathogenic *E. coli* (Appendix A).

### 3.3. Microbiome Composition Across Samples

Raw sequence data yielded 11,645,375 reads across all samples, with an average of 9414 ± 41,136 reads per sample (range: 2 to 694,039). After bioinformatic processing and quality controls—including removing ASVs that occurred in less than 10 samples and/or that had an abundance of five or less and that eliminated samples with fewer than 1000 reads—a total of 11,322,353 remained, with an average of 45,289 ± 82,333 reads per sample (range 1295 to 694,021). The ASVs detected in the negative controls, including known contaminants in DNA extraction kits, were identified and removed from the dataset based on a threshold of 100 or more reads in control samples. After filtering contaminants, 6,133,246 reads remained with an average of 24,381 ± 7077 forward/reverse reads per sample (range: 1295 to 108,934). Rarefaction analysis of fecal microbiome samples demonstrated sufficient sequencing depth, with curves reaching plateau at all read depths and when truncated to the minimum observed depth of 1000 reads, indicating comprehensive capture of species diversity within samples even after quality control (Figure 1).

### 3.4. Effects of Feed Sanitizer on Microbial Composition and Diversity of Sow Fecal Samples

For sows, 4,339,777 total sequences were obtained from fecal microbiome samples after bioinformatic processing, filtering, and decontamination. The depth range was 5386 to 108,934 reads and the average sequencing depth was 52,838 ± 15,042 reads per sample.

Bacterial alpha diversity analyses revealed no significant differences in Shannon Diversity indices from gestation to weaning in sows fed Control (Appendix A). However, richness expressed as the rarefied number of ASVs and the Chao1 index show both a trend (*p* = 0.06) and a significant (*p* = 0.01) reduction in bacterial diversity at weaning compared with gestation only in Control sows (Appendix A). For Treatment sows, no differences in alpha diversity were found from gestation to weaning for any of the indices aforementioned (Appendix A–F).

PERMANOVA analysis of Bray–Curtis distances visualized in PCoAs demonstrated significant compositional differences from gestation to weaning for both Control (*f* = 23.854, *r*^2^ = 0.271, *p* = 0.001; Figure 2A) and sanitizer diets (*f* = 13.391, *r*^2^ = 0.203, *p* = 0.001; Figure 2C). For Control diets only, PERMANOVA also revealed the significant effects of the sow cohort group for both weighted and unweighted distances, while parity and its interactions showed no significant effects (Appendix A).

A detailed examination of specific taxonomic changes across time within each dietary treatment (Figure 2B,D) was conducted using MaAsLin (version 2) to determine statistically significant shifts in bacterial ASVs along with its effect sizes, with a false discovery rate (FDR) adjustment threshold of *q* ≤ 0.005. The ASV that had the largest effect size discriminating day 80 of gestation from weaning in the Control group belongs to *Lactobacillus* sp. (*p* = 1.139 × 10^−11^), which distinguished the lactation stage. In contrast, an unknown ASV from the Lactobacillaceaes order (*p* = 8.057 × 10^−8^; Figure 2B) was the most depleted throughout gestation while *Mollicutes* RF39 (*p* = 9.595 × 10^−8^; Figure 2D) was the most depleted. In Figure 2B,D, bold fonts show taxa that uniquely fluctuated from gestation to weaning for each treatment group, which indicates that the main bacterial communities influenced by the physiological changes that take place across the gestation period are largely different for sows fed the Control and feed sanitizer diets. For sows under feed sanitizer treatment, an unknown *Peptoniphillus* sp. ASV was the most enriched throughout gestation (largest effect size, *p =* 5.422 × 10^−6^) transitioning to *Mollicutes* RF39 (*p* = 9.595 × 10^−8^; Figure 2D) at weaning.

PICRUSt was used for predicting the pathways associated with the taxonomic fluctuations aforementioned, which revealed distinct metabolic pathways enriched and depleted for each group through the gestation period. The five most significant pathways per diet are presented in Appendix A. Throughout the gestation period, Control sows showed a reduced abundance of predicted pathways related to vitamin B metabolism and heme synthesis, including thiamine diphosphate biosynthesis II, CMP-pseudaminate biosynthesis, pyridoxal 5′-pseudaminate biosynthesis, and two distinct heme b biosynthesis pathways (*p* < 0.05; Appendix A). In contrast, pathways associated with central metabolism were increased, including aromatic biogenic amine degradation, TCA cycle VI, sucrose degradation IV, L-isoleucine biosynthesis IV, and D-galactose degradation-I.

Sows fed the Treatment diet showed a reduced abundance of pathways for pyrimidine deoxyribonucleotides de novo biosynthesis II, norspermidine biosynthesis, flavin biosynthesis I, coenzyme A biosynthesis I, and trans, octa-cis decaprenyl phosphate biosynthesis throughout gestation, but showed an increase in pathways associated with complex carbohydrate processing and aromatic compound metabolism, UDP-glucose-derived O-antigen building blocks biosynthesis, two distinct toluene degradation pathways, dTDP-N-acetylthomosamine biosynthesis, and sulfur oxidation.

After exposure to treatment diets through the gestation period (at weaning), alpha diversity analyses revealed no significant differences in Shannon Diversity indices, Chao1 index, or richness between sows fed Control and sanitizer treatment diets (Appendix A). However, significant differences in gut microbiome composition were detected based on weighted Bray–Curtis distances (PERMANOVA, *f* = 3.230, *r*^2^ = 0.060, *p* = 0.001) and differences in ordination scores along PCo 3; further, because microbiome composition was not significantly different between the Control and Treatment groups at day 80 of gestation (PERMANOVA, *f* = 0.692, *r*^2^ = 0.011, *p* = 0.711), we attribute the microbiome compositional differences observed at weaning to the exposure to feed sanitizer throughout the gestation period (Figure 3A,B). The interaction between the treatment and parity groups was significant (PERMANOVA, *f* = 2139.0, *r*^2^ = 1.655, *p* = 0.002), while the effect of the parity group alone was not (Appendix A). At weaning, discriminant taxa with the greatest effect sizes were identified, with Control sows characterized by higher abundances of *Lactobacillus reuteri* and *Lactobacillus ruminis* (*p* = 1.473 × 10^−5^; Figure 3C,D), and sows fed the feed sanitizer by an increased abundance of *Sarcina* sp. (*p* = 9.167 × 10^−8^; Figure 3C,E).

Furthermore, metagenomic prediction analyses via PICRUSt revealed distinct metabolic pathways at weaning between Control sows and Treatment sows. Sows fed the Treatment diet had an increased abundance of pathways related to thiamine I and thiamine II diphosphate synthesis, and pyruvate to propanoate fermentation. Control sows showed a greater abundance of predicted pathways associated with glycolysis, sucrose degradation, bifidobacterium carbohydrate metabolism, and mevalonate production (Appendix A).

### 3.5. Effects of Feed Sanitizer on Microbial Composition and Diversity of Piglet Fecal Samples

For piglets, 1,793,469 total sequences were obtained from the fecal microbiome samples after bioinformatic processing and filtering for potential contaminants. The depth range was 1295 to 84,839 reads and the average sequencing depth was 52,838 reads per sample. Six samples were not considered further for bioinformatic analysis due to insufficient sequencing depth.

No alpha diversity differences were evident between the piglets from Control and Treatment fed sows, according to the Shannon Diversity Index, rarefied richness, and Chao1 richness (Appendix A). Bray–Curtis distance ordination analysis revealed distinct compositional clustering among piglet fecal samples; however, these clusters were not attributable to maternal dietary treatment in the PERMANOVA analyses (Figure 4A; *f* = 1.058, *r*^2^ = 0.022, *p* = 0.317), and instead they largely reflect the farrowing group (ellipses).

To elucidate the source of the compositional variation in Figure 4A, the Bray–Curtis distance ordination was reexamined based on the farrowing groups (Group 1—purple, Group 2—yellow, with corresponding purple and yellow ellipses). This analysis demonstrated that the observed compositional differences were significantly associated with farrowing groups (*f* = 11.354, *r*^2^ = 0.231, *p* = 0.001; Figure 4B) as opposed to sow treatment.

To further mine for potential microbiome community differences between pigs farrowed from each sow treatment, both alpha and beta diversity analyses were conducted separately for each farrowing group. The Bray–Curtis PCoA plot of piglets from Group 1 showed no significant differences between pigs from each dietary treatment (Figure 4C). Shannon Diversity, rarefied richness, and Chao1 richness index plots of piglets from Group 1 revealed no significant treatment effects (Appendix A). The Bray–Curtis PCoA plot of piglets from Group 2 showed no significant differences between pigs from each dietary treatment either (Figure 4D), across Shannon Diversity, rarefied richness, and Chao1 richness indexes (Appendix A). PICRUSt and MaAsLin analysis revealed no discriminant taxa or pathways between piglets farrowed from Treatment sows.

## 4. Discussion

This study evaluated how sanitizer supplementation in sow diets during late gestation and lactation affects production parameters and gut microbiome composition. Our findings show that sanitizer supplementation induced modest changes in the maternal gut microbiome while maintaining sow and litter performance.

### 4.1. Feed Sanitizer Level

Final concentrations of feed sanitizer were determined to ensure adequate levels and that treatments were imposed as designed. Initial levels of feed sanitizer were higher than designed in lactation diets. Due to the higher than designed levels of feed sanitizer in the lactation diets, feed was resampled. There was no mixing of the feed between sampling. After resampling, lactation diets ranged from 5.43 to 6.85 kg/metric ton which was deemed acceptable for the continuation of the study. Control diets were not affected as they did not contain any reported concentrations of feed sanitizer.

We speculate that the decrease in feed sanitizer concentrations from initial sampling to resampling is attributable to off-gassing, a process where volatile compounds like formaldehyde gradually evaporate from treated materials into the surrounding air. No research has been reported regarding formaldehyde off-gassing in swine feed; however, formaldehyde off-gassing is commonly observed in wood-based products assembled with urea–formaldehyde resins [24]. Researchers have documented that increased ventilation rates reduce concentrations of formaldehyde in indoor air [25,26]. However, forced air exchanges do not occur in the head space of feed storage bins. Limited passive air exchange between the head space and outdoor air might occur through roof vents or lids that are not airtight. The impact of this passive air exchange on the formaldehyde concentration of head space air and feed is not known. We did not measure the formaldehyde concentration in the head space air of feed storage bins used in this experiment. Off-gassing likely explains why our feed sanitizer concentrations decreased from initial levels to lower levels upon resampling, as the volatile formaldehyde compounds gradually released from the feed matrix over time. This process was also likely influenced by factors such as storage duration, temperature, humidity, and air circulation. In our study, we speculate that there would be further off-gassing of the swine feed if storage time increased.

### 4.2. Feed Sanitizer Has Minor Effects on Sow and Piglet Performance

The lack of significant differences in sow feed intake, body condition scores, and most reproductive parameters between treatment groups suggests that feed sanitizer supplementation can be implemented without compromising sow or litter performance. By random chance, sows fed feed sanitizer had lower body weight than Control sows on day 80 of gestation when dietary treatments were initiated. We accounted for this difference by using initial sow body weight as a covariate for subsequent measurements of sow weight. After statistically adjusting for differences in initial sow weight, sows fed feed sanitizer remained lighter than Control sows throughout the experiment. However, these differences in sow body weight did not translate to differences in piglet performance or subsequent reproductive performance.

We expected that the sanitized feed would potentially improve sow performance because it would spare sows the significant energy costs associated with immune system activation from potential pathogen exposure in feed. Sauber et al. [27] found that when sows underwent high immune system activation (induced by a subcutaneous lipopolysaccharide administration), compared to low or no immune system activation, they experienced a 10% reduction in feed intake, 12% reduction in litter weight gains, and 11% reduction in milk energy yield. These researchers determined that when animals encounter antigens, their immune system becomes activated, triggering cytokine release that alters metabolism by increasing the basal metabolic rate needed to maintain immune responses, and suppressing appetite in the process. This immune activation creates a substantial energy burden that diverts the energy required for milk production toward immune responses [27].

This competition for energy is particularly critical for lactating sows because lactation represents the highest energy demand during the reproductive cycle of a female mammal. In rodents, caloric intake during lactation can be 100% greater than during non-reproductive days [28,29]. Similarly, the daily metabolizable energy requirement of lactating sows is over 2.5 times greater than the daily ME requirement during gestation [30]. In short, lactation and milk production demand a great amount of nutrients. If sows cannot meet energy demands through diet alone, they will likely either reduce milk output, mobilize nutrients from maternal tissues, or a combination of both [31].

However, sows in the current experiment were not facing pathogen exposures that would result in significant immune activation. Evaluating feed sanitizers under sanitary conditions represents a key limitation of this study, as the benefits of a feed sanitizer may only become apparent when animals are actively challenged with feed-borne pathogens. We speculate that if sow performances were evaluated when diets contained significant pathogen loads, there might be more significant performance differences between treatment groups, as sanitized feed would theoretically reduce the pathogen infection of sows consuming sanitized feed. Without such challenge testing, researchers cannot make definitive conclusions about the efficacy of feed sanitizers for improving sow performance under pathogen pressure. This limitation in our research highlights the need for future studies that incorporate challenge models to fully evaluate the potential benefits of feed sanitization in commercial production settings.

The use of formaldehyde-based feed sanitizers with pigs from weaning to market weight elicited variable performance responses. When researchers treated spray-dried animal plasma—a single ingredient—with 0.003 kg formaldehyde/tonne of feed, nursery pig performance improved. However, treating entire diets with 0.003 kg formaldehyde/tonne of feed (3 mg/kg) failed to enhance growth performance [11]. Much higher levels of formaldehyde treatment of the entire diet (1000 mg/kg diet) reduced the average daily gain and final body weight of nursery pigs compared to nursery pigs who consumed untreated diets [13]. A dietary formaldehyde–propionic acid blend including up to 3.0 g/kg which supplied up to 990 mg formaldehyde/kg feed did not affect growing pig performance [7]. Previous investigations [7,11,13] employed lower inclusion rates of feed sanitizer compared to the rate used in the current study.

None of the aforementioned studies were conducted in a high dietary pathogen model. These varied responses suggest that the effects may depend on the formaldehyde application method (individual ingredients versus complete diets), inclusion level, pathogen load in the feed, and life stage of the pig.

Most of the research on formaldehyde-based feed sanitizers in swine has been conducted in nursery or growing-finishing pigs rather than mature sows, which presents limitations when applying these findings to breeding stock. Growing pigs differ from sows in their physiological status—particularly in microbiome and immune system maturation. While in the nursery stage, pigs are experiencing the significant stress of developing their immune systems as serum protein levels do not reach mature concentrations until 21–24 weeks of age [32]. In contrast, sows have fully established immune systems and face entirely different physiological challenges related to reproduction, including hormonal fluctuations through estrous cycles, pregnancy, and lactation [33]. These life-stage distinctions are critical when interpreting the effects of feed sanitizers on performance and microbiome composition.

The stressors affecting growing pigs and sows diverge substantially—growing pigs contend with new environments, unfamiliar odors, and social reorganization triggering stress responses and inflammatory protein production [34,35], while sows experience metabolic and immunological challenges associated with milk production and reproductive demands [31,36]. This physiological context might explain why our results on sow and piglet performance may differ from previous research focused on younger animals. The observations of life stage support the idea that specific research is necessary to properly evaluate feed additives like formaldehyde-based sanitizers for sows. Our finding of minimal impacts on sow performance contributes new information about formaldehyde’s effects in breeding animals while generally aligning with previous research showing its limited effects on growth at recommended levels.

The high proportion of sows returning to estrus within 7 days post-weaning in both groups (98% Control, 94% feed sanitizer) indicates that supplementation did not interfere with the complex endocrine mechanisms required to initiate post-weaning estrus. This maintenance of reproductive efficiency is crucial for commercial production systems where consistent breeding performance is essential.

There was a low incidence of scouring among piglets observed in our study. We speculate that the low rate likely resulted from a high health status of the sow herd and our use of slotted cast iron (under sow) and plastic-coated woven wire (in piglet areas) in farrowing stalls. Additionally, the farm staff caring for the litters and sows maintained rigorous cleaning protocols, which included power washing farrowing rooms with hot water and disinfectants between farrowing groups.

### 4.3. Limited Differences in Sow Microbiome Attributed to Feed Sanitizer

Sow microbiome analyses revealed temporal dynamics in both experimental groups, with significant compositional shifts occurring between gestation and weaning. These changes likely reflect physiological (immune) and metabolic adaptations during the transition from gestation to lactation and the addition of feed sanitizer. Previous research [37] found similar temporal changes throughout gestation in a comprehensive analysis of gut microbiota changes across 12 time points during the perinatal period in sows. Their study demonstrated that the transition from prenatal to postnatal periods significantly influenced gut microbiota, creating clear differences in microbial diversity and composition between phases. They specifically found higher alpha diversity during lactation compared to gestation, and distinct clustering of microbial communities between gestation and lactation.

We similarly found differences in microbial diversity and composition between perinatal phases, but a significant decrease in alpha diversity from gestation to weaning was only detected for the Control sows. This observation indicates that feed sanitizer supplementation may be associated with less alpha diversity disruptions throughout the gestational period. Thus, feed sanitizer supplementation may modulate community composition while maintaining overall alpha diversity, which suggests that formaldehyde-based sanitizer does not disrupt community stability or complexity.

While the strong microbiome shifts across reproductive stages likely overshadowed more subtle sanitizer-related differences in our study, sows from each treatment group experienced different taxonomic and functional changes throughout pregnancy and weaning. For example, in Control sows, the dynamic shift in ASVs related to Lactobacillales through gestation aligns with known immune and metabolic transitions during this perinatal period. This observation partially aligns with observations that the abundance of Lactobacillales typically increases from gestation to lactation in sows [38]. Other studies have found that *Lactobacillus* spp. vary in the sow fecal microbiome from 100 days of gestation to 21 days after parturition [37,39]. Given the immunomodulatory properties reported for members of this order, this shift is hypothesized to support maternal health during this metabolically and immune demanding reproductive phase.

Additionally, the significant increase in *Lactobacillus* at weaning for the Control group coincides with the critical adaptations needed during lactation; because Lactobacillus makes up part of maternal milk, its stimulation at the gut level could support the potential enrichment in milk through the entero-mammary pathway [40]. Lactobacilli have also been correlated positively with secondary bile acid biosynthesis [41]; these microbial metabolites play a vital role in maintaining gut microbiota balance and immune function and supporting lipid metabolism during the end stages of gestation [37].

ASVs associated with the *Lactobacillales* order were also identified in the sanitized diet treatment group, distinguishing gestational changes, but the effect size was not as prominent as that seen on the Control sows. This observation indicates that the bacteria which are mostly fluctuating throughout gestation and that have important immune and metabolic roles may be affected by feed sanitation.

For example, the taxa that mostly fluctuated in Treatment sows throughout gestation were *Peptoniphilus* spp., predominating at day 80 of gestation. This Gram-positive anaerobic genus that ferments peptones (protein hydrolysates) has been associated with potential benefits including improved gut barrier function, reduced intestinal inflammation, and increased production of butyrate and acetate [42]. Short-chain fatty acid production is critical to sows during late gestation when energy demands are the greatest of any period of pregnancy. However, both the Control and Treatment groups show variations in different taxa equally associated with fermentation roles from gestation to weaning.

Thus, formaldehyde-based feed sanitizer supplementation may stimulate different taxa across gestation compared to Control sows, but such taxa may adapt to the same metabolic or energetic constraints faced by the sow during this period. However, the fact that the top predicted metabolic pathways that vary during this period in Treatment and Control are also different warrants further investigation. Such is the case of the increased abundance of pathways associated with carbohydrate metabolism in Control sows and the higher abundance of pathways involved in xenobiotic degradation in Treatment sows at weaning.

During lactation, Control sows increased the activity of energy metabolism pathways, particularly in the TCA cycle, while enhancing carbohydrate utilization through sucrose and galactose degradation. Lactogenesis requires glucose and lactose, which are sourced from high energy diets and endogenously [43,44,45]. We speculate that these microbiome changes represent an adaptive response that meets the heightened metabolic demands of milk production.

Treatment sows during lactation enhanced their capacity for the degradation of aromatic compounds and sulfur metabolism. The elevated toluene degradation pathways suggest an enhanced capacity to metabolize aromatic compounds, possibly including formaldehyde or its derivatives. The increased activity of sulfur oxidation pathways indicates an altered sulfur metabolism, which could affect amino acid synthesis and antioxidant defense systems which are important during the metabolically demanding lactation period.

Comparing samples collected at weaning only, when sows had already been exposed to the treatment diets, we observed a moderate but significantly different bacterial community composition between the Control and Treatment sows. *Sarcina* was detected as the taxon that most distinguished the latter. *Sarcina* spp. have been associated with gastrointestinal pathologies including mucosal hyperemia, hemorrhage in non-ruminants, and delayed gastric emptying [46]. However, *Sarcina* has also been associated with the production of potentially beneficial polyamines, like putrescine [47]. In this regard, reports demonstrated that dietary putrescine supplementation in weaned piglets results in beneficial effects, including enhanced intestinal development, improved immune function through increased lysozyme and immunoglobulin M levels, and reduced inflammatory responses [48]. However, current research shows no evidence that *Sarcina*, stimulated by a feed sanitizer, is actively producing putrescine or enhancing immunity.

Of note is the significantly reduced abundance of *Lactobacillus ruminis* and *Lactobacillus reuteri* in the fecal microbiome of sows from the sanitizer group compared to Control sows at weaning [49,50,51]. This observation is in line with the hypothesis raised above, that feed sanitizer is associated with the suppression of the taxa typically involved in immunomodulation. The implications of these findings for sow health and productivity are unclear, especially in the hyper-sanitized environment characterizing this experiment. However, a key question to pursue can focus on whether feed sanitizer spares the sow microbiome from investing in immunomodulation.

The functional prediction profiles of both treatment groups when compared at weaning do not reveal many clues on this matter. However, it is noteworthy that the gut predicted pathways of sows under feed sanitizer through gestation show profiles that may denote more active fermentation of complex carbohydrates at weaning (e.g., pyruvate fermentation to propanoate and incomplete reductive TCA cycle), while Control sows exhibit predicted pathways more associated with metabolizing simple sugars (e.g., sucrose degradation, glycolysis; Appendix A). These observations are consistent with the increased abundance of Lactobacillus species in the Control sows at weaning.

### 4.4. Lack of Differences in Piglet Microbiome Analysis

Piglet microbiome analysis revealed clear compositional clusters primarily linked to farrowing groups rather than maternal dietary treatment. We propose that feed sanitizer in sow diets has a negligible effect on the development of the piglet’s gut microbiome compared to other factors such as microbial succession from farrowing to weaning, consumption of milk, and the environmental conditions early in life. These factors may influence early-life microbiome development in a more powerful way than modifications of the maternal diet.

## 5. Conclusions

This study demonstrated that formaldehyde-based feed sanitizer supplementation in sow diets during late gestation and lactation maintains reproductive performance while inducing modest alterations in the maternal gut microbiome. Both Control and sanitizer-supplemented sows exhibited significant temporal shifts in their microbial communities from gestation to weaning, likely reflecting the physiological adaptation to lactation rather than dietary intervention. However, this study highlights that the taxa involved in the changes that take place throughout gestation may be associated with the introduction of feed sanitizer. Although discriminant taxa emerged between experimental groups at weaning, these changes occurred without major disruption of the overall community composition (especially alpha diversity), suggesting subtle microbiome modulation rather than comprehensive restructuring.

The findings present several practical implications for swine production. Feed sanitizer supplementation can be implemented without significant negative impacts on sow performance or reproductive function, making it a viable option for producers seeking to improve feed safety. While the supplement does not drastically influence the gut microbial ecology, the observed microbial changes in diversity and taxa composition remain subtle and do not disrupt community stability. Additionally, we speculate that the limited transmission of treatment effects to piglet microbiomes indicates that the maternal dietary intervention implemented in this study has less influence on offspring gut colonization than environmental factors, suggesting that producers should focus equally on environmental management practices alongside dietary interventions to optimize piglet health outcomes.

Our findings contribute valuable insights to the limited body of research on feed sanitizers in reproducing sows, though results should be interpreted within the context of our controlled research environment using a specific sanitizer formulation and inclusion rate. Future research should explore the functional implications of the observed taxonomic shifts and examine performance under commercial conditions where animals face various health challenges and environmental stressors.

## Figures and Tables

**Figure 1 animals-15-03618-f001:**
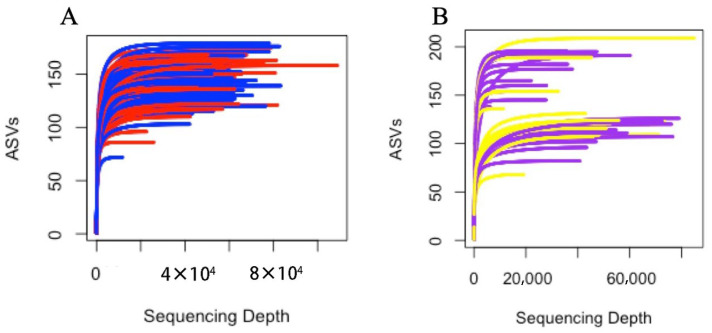
Rarefaction curves for the samples collected from sows and piglets. (**A**) is the curve for all 118 samples collected from sows, colored based on diets (blue = Control diets, *n* = 63 samples; red = Treatment diets, *n* = 55 samples); (**B**) is the curve for all 39 samples collected from piglets, (purple = Control diets, *n* = 22 samples; yellow = feed sanitizer diets, *n* = 17 samples); ASVs = amplicon sequence variants.

**Figure 2 animals-15-03618-f002:**
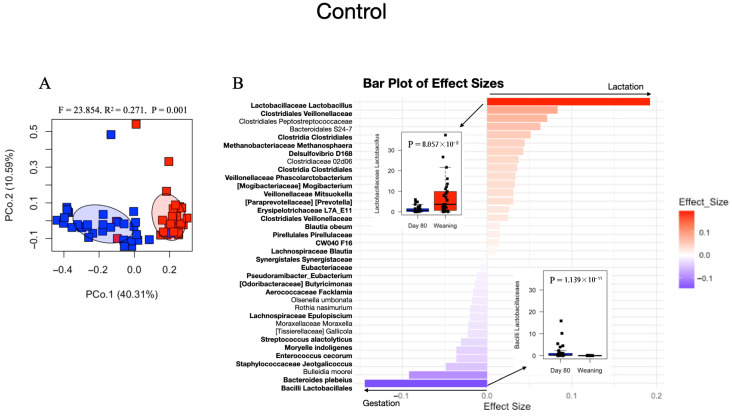
Beta diversity and discriminant taxa distinguishing gestation from weaning for Control (*n* = 63) and Treatment diets (*n* = 55) sows. (**A**) Principal coordinate analysis (PCoA) plot based on Bray–Curtis dissimilarity for fecal samples collected from sows fed Control diets. (**B**) Effect sizes for bacterial taxa (ASVs) that significantly fluctuated from gestation to weaning for Control sows, as determined by MaAsLin analysis (*q*-value ≤ 0.005; red = taxa that increased significantly during lactation, blue = taxa that were reduced). An example of the top enriched taxon at lactation for the Control sows is shown in the inset boxplot of an unknown Lactobacillus ASV, which showed the largest effect size. In contrast, an unknown Lactobacillaceaes ASV was the most depleted taxon in Control sows throughout gestation. (**C**) Principal coordinate analysis (PCoA) plot based on Bray–Curtis dissimilarity for sows fed sanitized diets. (**D**) Effect sizes of bacterial taxa that significantly fluctuated from gestation to weaning, as determined by MaAsLin analysis (*q*-value ≤ 0.005; red = taxa significant during lactation, blue taxa significant during gestation). The inset shows a boxplot of an unknown *Mollicutes* RF39 ASV, which showed the largest effect size, increasing significantly in abundance at lactation in sows fed the feed sanitizer. In contrast, an unknown *Peptoniphilus* sp. ASV was the most significantly depleted taxon in this group (e.g., it showed the lowest effect size throughout gestation). Bold labels show taxa that uniquely fluctuated within each group from gestation to weaning.

**Figure 3 animals-15-03618-f003:**
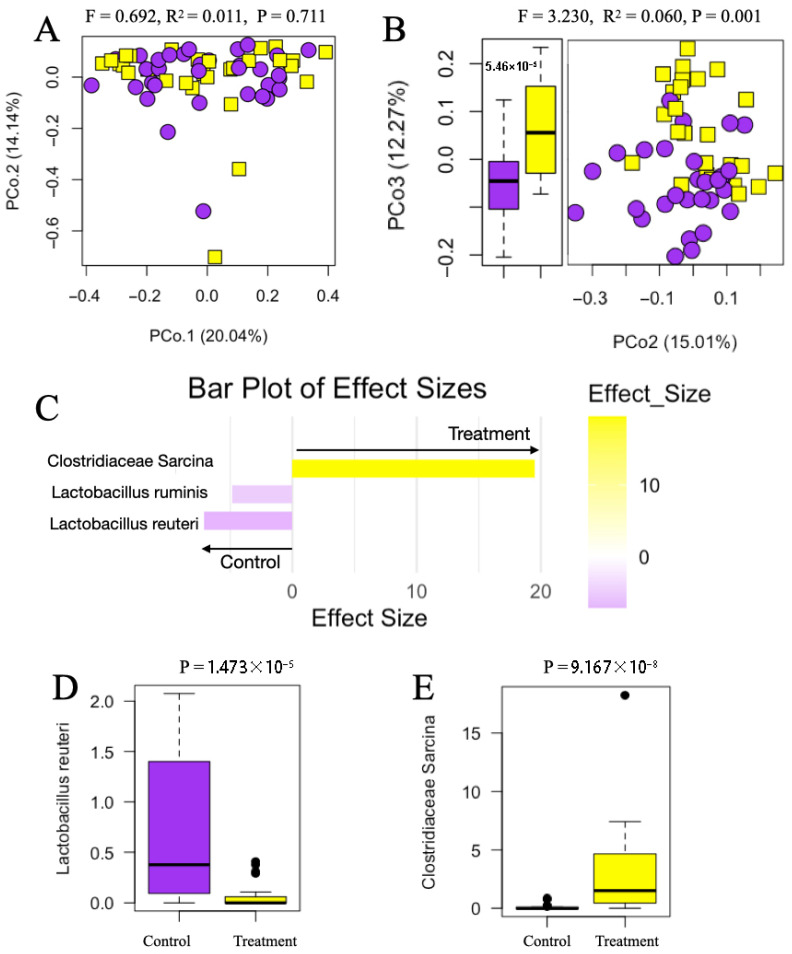
Gut microbiome differences between Treatment (*n* = 55) and Control sows (*n* = 63) at gestation. Across all panels, Treatment samples are shown in yellow and Control samples are shown in purple. (**A**) Principal coordinate analysis (PCoA) plot based on Bray–Curtis dissimilarity comparing the two groups before treatments were imposed at day 80 of gestation. (**B**) Principal coordinate analysis (PCoA) plot based on Bray–Curtis dissimilarity (vectors 2 and 3) for fecal samples collected from sows at weaning, across both diets. To the left of the PCoA is a boxplot showing a significant difference in the distribution of ordination scores by treatment group along PCo3. (**C**) Effect sizes of significant bacteria distinguishing Treatment from Control sows at weaning, as determined by MaAsLin analysis ((*p* ≤ 0.05, *q* ≤ 0.05); yellow = taxa significant in Control sows, purple taxa significant in Treatment sows). (**D**) Boxplot of *Lactobacillus reuteri*, identified as the taxon with the largest effect size (most abundant taxa) within sows fed Control diets at weaning. (**E**) Boxplot of *Sarcina* sp. identified as the taxon that most distinguished sows fed Treatment diets.

**Figure 4 animals-15-03618-f004:**
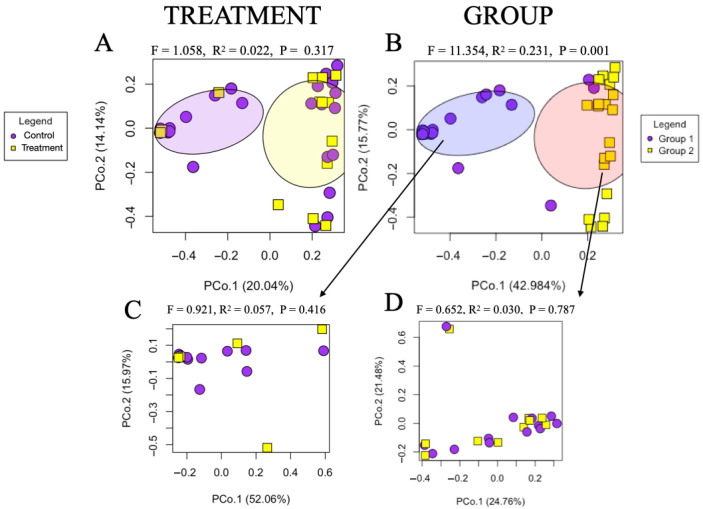
Gut microbiome composition of piglets farrowed from Control or Treatment sows. (**A**) Principal coordinate analysis (PCoA) plot based on Bray–Curtis comparing piglets farrowed from sows fed each dietary treatment (Control*—*purple, *n* = 22; Treatment*—*yellow, *n* = 17). Shaded ellipses show that farrowing group was the main driver of the separation observed along PCo.1. (**B**) Principal coordinate analysis (PCoA) plot based on Bray–Curtis dissimilarity confirmed significantly different microbiome composition in piglets from two different farrowing groups (red ellipse = Group 1, *n* = 18; blue ellipse = Group 2, *n* = 21). (**C**) Principal coordinate analysis (PCoA) plot based on Bray–Curtis dissimilarity for piglets in Group 1 (*n* = 16) across dietary treatments. (**D**) Principal coordinate analysis (PCoA) plot based on Bray–Curtis dissimilarity for piglets in Group 2 (*n* = 23) across dietary treatments.

**Table 1 animals-15-03618-t001:** Composition of experimental diets fed during gestation and lactation (% as fed).

	Gestation		Lactation	
Item	Control	Treatment	Control	Treatment
Corn, yellow dent	58.72	58.17	66.39	65.84
Soybean meal, dehull, solv. extr. (46% CP)	8.00	8.00	18.00	18.00
DDGS ^1^, >6 and <9% oil	15.00	15.00	10.00	10.00
CWG ^2^	2.00	2.00	1.50	1.50
Limestone	1.00	1.00	1.40	1.40
Monocalcium phosphate	1.60	1.60	1.20	1.20
Soybean hulls	12.50	12.50	-.-	-.-
Salt	0.35	0.35	0.50	0.50
Swine breeder pmx ^3^	0.50	0.50	0.50	0.50
L-Lysine HCl (78%)	0.30	0.30	0.40	0.40
L-Threonine (98.5%)	0.03	0.03	0.08	0.08
L-Tryptophan (98%)	-.-	-.-	0.01	0.01
L-Valine (97.5%)	-.-	-.-	0.02	0.02
Feed sanitizer ^4^	-.-	0.55	-.-	0.55
Total	100.0	100.0	100.0	100.0
	** Calculated ** ** Nutrient Content, % **			
	** Control Gestation **		** Control Lactation **	
NE, kcal/kg	2365		2494	
Ca%	0.75		0.89	
P, total%	0.66		0.66	
Na%	0.21		0.27	
STTD ^5^ P%	0.45		0.42	
Crude protein	14.35		17.22	
Crude fiber	7.3		2.9	
SID ^6^ Arg	0.66		0.90	
SID His	0.32		0.40	
SID Ile	0.44		0.57	
SID Leu	1.22		1.40	
SID Lys	0.69		0.98	
SID Met	0.22		0.25	
SID Met + Cys	0.42		0.49	
SID Phe	0.57		0.71	
SID Thr	0.42		0.58	
SID Trp	0.10		0.16	
SID Val	0.53		0.68	
SID Lys/NE, g/Mcal	2.90		3.93	

^1^ DDGS = Dried distiller’s grains with solubles. ^2^ Choice white grease. ^3^ Swine breeder premix was supplied by Form-A-Feed Services, Inc., Stewart, MN, USA. Mineral and vitamin mixture supplied per kilogram of diets: 18.5 mg of Cu (as CuSO_4_); 112.5 mg of Fe (as FeSO_4_·7H_2_O); 45 mg of Mn (as MnO); 122.5 mg of Zn (as ZnO); 1.0 mg of I (as Ca(IO_3_)_2_); 0.30 mg of Se (as Na_2_SeO_3_); 11,023 IU of vitamin A; 1477 IU of vitamin D_3_; 88 IU of vitamin E; 13.33 mg of vitamin K_3_; 6.47 mg of thiamin; 9.9 mg of riboflavin; 52.25 mg of nicotinic acid; 38.5 mg of D-pantothenic acid; 3.02 mg of pyridoxine; 0.05 mg of vitamin B_12_; 2.42 mg of folic acid; 0.33 mg of biotin; and 503.8 mg of choline. ^4^ The feed sanitizer in this study was Termin-8 (Anitox Corp., Lawrenceville, GA, USA), a formaldehyde and propionic-based feed sanitizer. ^5^ Standardized total tract digestible. ^6^ Standardized ileal digestible.

**Table 2 animals-15-03618-t002:** Summary of feed sanitizer concentration in experimental diets.

Date Sampled	Diet Sampled	Expected Dose (kg/metric ton)	Actual Dose Recovered (kg/metric ton)
21/8/23	Gestation batch #1	5.5	6.22
20/9/23	Lactation batch #1 ^1^	5.5	7.86
28/9/23	Lactation batch #1 (resample) ^2^	5.5	6.85
28/9/23	Lactation batch #2 ^1^	5.5	7.57
9/10/23	Lactation batch #1 (resample) ^2^	5.5	5.43
9/10/23	Lactation batch #2 (resample) ^2^	5.5	6.63

^1^ Sample collected from feed truck on delivery to sow barn. ^2^ Sample collected from feed bin at sow barn.

**Table 3 animals-15-03618-t003:** Effect of feed sanitizer fed to sows in late gestation and lactation on sow performance.

	Diet			*p*-Value				
Trait	Control	Treatment ^1^	Pooled SEM ^2^	Trt ^3^	Time	Trt × Time ^4^	Initial BW ^5^	Parity
No. of sows	53	54	-	-	-	-	-	-
Parity	2.1	2.4	0.568	0.511	-	-	-	-
Lactation length, days	19.8	19.9	0.257	0.440				0.791
Feed intake, kg			1.518	0.960	<0.001	0.995	0.625	0.029
Wk 1	33.6	33.9						
Wk 2	53.0	52.5						
Wk 3	43.7	43.8						
Total lactation intake	130.7	130.3	4.169	0.934				0.019
Average daily feed intake, kg			0.326	0.811	<0.001	0.965	0.913	0.009
Wk 1	7.57	7.48						
Wk 2	7.57	7.48						
Wk 3 ^6^	7.60	7.50						
Overall (farrow to wean)	6.6	6.6	0.217	0.849				0.038
Sow lactation feed efficiency ^7^	1.1	1.2	0.070	0.076				0.402
Sow body weight, kg			1.770	0.002	<0.001	0.568	<0.001	<0.001
Pre-treatment	-	-						
Day 109 of gestation	244.2	240.2						
24 h post-farrow	231.1	228.1						
Weaning	228.9	222.3						
Sow body weight change, kg								
Gestation body weight change	19.2	16.2	1.442	0.218				0.624
Farrowing body weight change	−13.1	−12.0	1.293	0.550				0.562
Lactation body weight change	−1.8	−6.2	2.135	0.067				0.004
Sow backfat depth, mm			0.358	0.292	<0.001	0.964	<0.001	0.952
Day 80 of gestation	10.8	11.0						
Day 109	11.3	11.7						
24 h	11.0	11.4						
Weaning	9.7	9.8						
Sow caliper			0.281	0.345	<0.001	0.792	<0.001	0.194
Gestation	14.2	12.4						
Weaning	14.0	12.1						
Sow caliper change	−1.7	−2.0	0.285	0.489				<0.001

^1^ The feed sanitizer in this study was Termin-8 (Anitox Corp., Lawrenceville, GA, USA), a formaldehyde and propionic-based feed sanitizer. ^2^ Standard error of the mean. ^3^ Dietary treatment. ^4^ Dietary treatment by time interaction. ^5^ Body weight at day 80 of gestation, before sows were administered their respective diets. ^6^ Week 3 did not include 7 days for all sows but ranged from 3 to 7 days. ^7^ Total lactation feed intake, kg/(Sow body weight change, kg + Litter weight gain, kg).

**Table 4 animals-15-03618-t004:** Effect of feed sanitizer on distribution of weaning-to-estrus intervals.

	Diet		
Trait	Control	Treatment ^1^	*p*-Value
Total sows weaned	53	54	-.-
Weaning-to-estrus interval to 7 d	52 ^2^	51	0.317 ^3^
Weaning-to-estrus interval to 14 d	52	51	
Weaning-to-estrus interval to 21 d	52	51	

^1^ The feed sanitizer in this study was Termin-8 (Anitox Corp., Lawrenceville, GA, USA), a formaldehyde and propionic-based feed sanitizer. ^2^ Number of sows. ^3^
*p*-value for chi-square analysis.

**Table 5 animals-15-03618-t005:** Effect of feed sanitizer fed to sows in late gestation and lactation on litter performance.

	Diet			*p*-Value				
Trait	Control	Treatment ^1^	Pooled SEM ^2^	Trt ^3^	Time	Trt × Time ^4^	Initial BW ^5^	Parity
No. of litters	53	54						
Litter size, *n*			0.319	0.102	<0.001	0.739	0.234	0.001
Total born alive	14.1	14.8						
After cross-fostering	13.6	14.0						
Weaning	12.6	12.8						
Piglets per litter, *n*								
Total born	16.7	16.4	0.434	0.571				0.392
Mummies	0.8	0.5	0.137	0.120				0.528
Stillborn	1.8	1.2	0.283	0.048				0.142
Piglet mortality before cross-fostering	0.5	0.7	0.193	0.229				0.656
Piglet mortality after cross-fostering	0.6	0.8	0.219	0.593				0.195
Piglet mortality, %								
After cross-fostering ^6^	4.4	4.2	1.278	0.953				0.450
Litter weight, kg			0.923	0.987	<0.001	0.825	0.004	0.059
Total born alive	20.5	20.9						
Total after cross-fostering	20.0	20.3						
Weaned	78.4	77.8						
Litter weight, kg								
Mummies	0.3	0.2	0.071	0.325				0.979
Stillborn	2.2	1.2	0.347	0.008				0.259
Transferred off	2.1	1.8	0.585	0.381				0.610
Dead pigs before cross-fostering	1.2	1.8	0.439	0.145				0.192
Average piglet weight, kg								
Born alive	1.5	1.4	0.026	0.247				0.679
Stillborn	0.9	0.7	0.086	0.059				0.534
Dead pigs after cross-fostering ^7^	0.6	0.5	0.234	0.340				0.820
Litter weight gain, kg ^8^	58.4	57.4	1.314	0.574				0.408

^1^ The feed sanitizer in this study was Termin-8, (Anitox Corp., Lawrenceville, GA, USA), a formaldehyde and propionic-based feed sanitizer. ^2^ Standard error of the mean. ^3^ Dietary treatment. ^4^ Dietary treatment by time interaction. ^5^ Body weight at day 80 of gestation, before sows were administered their respective diets. ^6^ [(Dead pigs after cross-fostering/litter)/(Total piglets/litter after cross-fostering)] × 100. ^7^ Total weight of dead pigs after cross-fostering/litter)/(Number of dead pigs after cross-fostering/litter). ^8^ Total litter weight at weaning − Total litter weight after cross-fostering.

## Data Availability

The raw data supporting the conclusions of this article will be made available by the authors on request.

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
