# Peer review of "Effects of a Feed Sanitizer in Sow Diets on Sow and Piglet Performance"

_animals, 2025, doi:10.3390/ani15243618_

Round 1

Reviewer 1 Report

Comments and Suggestions for Authors

Effects of a feed sanitizer in sow diets on sow and piglet performance

Dear Authors,

The manuscript is interesting and quite well prepared. As described in conclusions during experiment no significant differences was determined in case of the performance, but in my opinion maybe better will be compare data from point of view the one-way ANCOVA (with two covariates and diet as an experimental factor) instead of the two-way ANCOVA (with two factors, interaction between them, and two covariates), that allows to find possible differences during different dates of measurements/periods of experiment. Study was compared on minimally 53 animals in each experimental group/treatment (replications in case of piglets), what provide the appropriate and high power of a test. In this case taking into consideration outliers in case of each group and for example total born alive piglets (Table 5) treatment vs control (respectively 14.8 and 14.1) should be rather significant. Sow caliper and comparison in gestation and weaning phase between diets (Table 3). Additionally in my opinion more references can be added in Discussion section, because there are several paragraphs, without any references.

Below I added some suggestions/comments helpful during revision process:

Line 18

Maybe better to use ‘their offspring’ than ‘their babies’.

Line 38

‘Feed sanitizer supplementation had no significant effect (p>0.05)…’, can be added.

Lines 40-506

p-value instead of P-value must be used (samples gathered from 107 sows and x*107 piglets).

The same in case of r2 and f.

Lines 51-779

Text must be justified.

Line 68

Three references in a row: [9-11].

Lines 95-742

Second dot in case of subsections and sub-subsections headers must be added.

2.1. Animals, Housing, and Treatments

2.2.1. Sows

Line 109

ad libitum (with italics)

Line 113

Table 1

Monocalcium phosphate, entire name can be specified.

Kind of isomer can be also added in case of amino acids L-Threonine, L-Tryptophan and L-Valine. Content of pure amino acid in ingredient can be also determined in brackets, ie. L-Lysine HCl (78%),...

Crude protein content in SBM can be also given.

Line 137

Probably, between 08:00 and 10:00 (am/24h?).

Lines 249-260

Maybe is possible to add in this part equation describing model with two covariates and interaction between experimental factors with description?

Lines 258-259

p-values

Lines 309-336

p-value in mentioned line 309 and header of Tables 3-5 must be added.

Line 315

Table 3

Top border must be added.

Maybe is possible to compare FI, ADFI also between control and experimental treatment, even from point of view one-factorial covariance with two covariates?

The same in case of Pre-treatment for Day 109 of gestation, 24h post farrow and Weaning comparison between.

and so on… to Sow caliper.

Line 335

Table 5

Top border must be added.

Maybe is possible the same as in case Table 3 compare all parameters in case od diet factor (with two covariates)?

Lines 339-340

  1. coli (binominal nomenclature with italics).

Line 410

Lactobacillus sp. or spp?

Lines 411-456

Better form which can be used is p = 1.139 × 10-11.

Line 418

Peptoniphilus sp. or spp?

Lines 453 and 468

Sarcina sp. or spp?

Lines 476-477

One row of space must be added between.

Lines 551-560

Maybe is possible to compare this paragraph with other study from point of view sows’ performance, BCS or reproductive parameters?

Line 554

Body weight.

Lines 581-592

In case of this paragraph is possible to confront obtained results with other studies.

Lines 654-660, 693-700 and 707-712

The same as in lines 581-592.

Lines 668-673

Lactobacillus sp. or spp?

Line 418

Genus: Peptoniphilus sp. or spp?

Lines 716-723

Genus or Sarcina spp.

Line 725

Lactobacillus ruminis and Lactobacillus reuteri (italics required).

Lines 743-749

The same as in lines 581-592.

Lines 818-902

Abbreviation/s of Journal’s name must be used.

Line 904

Sci. Rep.

Lines 909, 913 and 919

Abbreviation/s of Journal’s name must be used.

Line 911, 935, 942 and 947

Dot/point on the end of each abbreviation of Journal’s name required.

Lines 934-953

Information about purpose of references [A] – [I] can be added in header.

Author Response

The manuscript is interesting and quite well prepared. As described in conclusions during experiment no significant differences was determined in case of the performance, but in my opinion maybe better will be compare data from point of view the one-way ANCOVA (with two covariates and diet as an experimental factor) instead of the two way ANCOVA (with two factors, interaction between them, and two covariates), that allows to find possible differences during different dates of measurements/periods of experiment. Study was compared on minimally 53 animals in each experimental group/treatment (replications in case of piglets), what provide the appropriate and high power of a test. In this case taking into consideration outliers in case of each group and for example total born alive piglets (Table 5) treatment vs control (respectively 14.8 and 14.1) should be rather significant.  Sow caliper and comparison in gestation and weaning phase between diets (Table 3). Additionally in my opinion more references can be added in Discussion section, because there are several paragraphs,

without any references. 

Response:  Thank you for the kind words.  We disagree with advice to change our statistical approach to analyzing our data.  If we understand the reviewer’s suggestion correctly, this reviewer’s suggestion ignores the repeated nature of the data.  Data were collected from the same set of sows and piglets at various times during the experiment.  This design calls for a repeated measures analysis (which we did) because of the inherent correlation of errors among observations that are collected sequentially on the same experimental units (sows and litters). 

We are not clear what references should be added to the discussion.  In our opinion, we have cited all the pertinent literature related to providing feed sanitizers to sows.  Adding references just for the sake of having citations in the text does not seem to add to our story. 

Below I added some suggestions/comments helpful during revision process:

Line 18

Maybe better to use ‘their offspring’ than ‘their babies’.

Response:  Suggested change made

Line 38

‘Feed sanitizer supplementation had no significant effect (p>0.05)…’,

can be added.

Response:  This is a style change that we do not agree with.  If we include such a designator here, we need to do it throughout the paper to be consistent.  We state our critical P value is 0.05 so it seems redundant to include the same information throughout the paper.  This unnecessarily clutters the paper.  No change made.

Lines 40-506

p-value instead of P-value must be used (samples gathered from 107

sows and x*107 piglets).

The same in case of r and f.

Response:  Suggested style changes made throughout the manuscript

Lines 51-779

Text must be justified.

Response:  We are confused by this comment.  We used the template provided on the website to prepare this submission.  This template does not have text right justified.  We are happy to do so if the editor asks for such. 

Line 68

Three references in a row: [9-11].

Response:  Suggested change made.

Lines 95-742

Second dot in case of subsections and sub-subsections headers must

be added.

2.1. Animals, Housing, and Treatments

2.2.1. Sows

Response:  Suggested change made

Line 109

ad libitum (with italics)

Response:  Suggested change made.

Line 113

Table 1

Monocalcium phosphate, entire name can be specified.

Kind of isomer can be also added in case of amino acids L-Threonine,

L-Tryptophan and L-Valine. Content of pure amino acid in ingredient

can be also determined in brackets, ie. L-Lysine HCl (78%),...

Crude protein content in SBM can be also given.

Response:  Suggested changes made

Line 137

Probably, between 08:00 and 10:00 (am/24h?).

Response:  We expressed this as military time so there is no need for am and pm designations.  No change made.

Lines 249-260

Maybe is possible to add in this part equation describing model with two covariates and interaction between experimental factors with description?

Response:  See response above.  We think our current statistical analysis is correct. 

Lines 258-259

p-values

Response:  As indicated above, all these p values have been reformatted.

Lines 309-336

p-value in mentioned line 309 and header of Tables 3-5 must be added.

Response:  Formatting of these p values has been corrected.

Line 315

Table 3

Top border must be added.

Response:  Top borders have been added to all tables.

Maybe is possible to compare FI, ADFI also between control and experimental treatment, even from point of view one-factorial covariance with two covariates?

The same in case of Pre-treatment for Day 109 of gestation, 24h post farrow and Weaning comparison between. and so on… to Sow caliper.

Response:  We do not understand this comment.  We have compared control and experimental treatments for FI and ADFI and other response variables.

Line 335

Table 5

Top border must be added.

Maybe is possible the same as in case Table 3 compare all parameters in case od diet factor (with two covariates)?

Response:  See responses above for Table 3.

Lines 339-340

  1. coli (binominal nomenclature with italics).

Response:  Suggested change made

Line 410

Lactobacillus sp. or spp?

Response:  Correction made.

Lines 411-456

Better form which can be used is p = 1.139 × 10 .

Response:  Both forms are correct and the figures already reflect our notation so we would like to keep the current form. 

Line 418

Peptoniphilus sp. or spp?

Response:  Correction made.

Lines 453 and 468

Sarcina sp. or spp?

Response:  Correction made.

Lines 476-477

One row of space must be added between.

Response:  Blank line has been added.

Lines 551-560

Maybe is possible to compare this paragraph with other study from point of view sows’ performance, BCS or reproductive parameters?

Response:  As was pointed out in the introduction, we found no other reports of formaldehyde-based feed sanitizers being fed to sows in the scientific literature.  Hence there is nothing to compare with.  No change made.

Line 554

Body weight.

Response:  Requested change made.

Lines 581-592

In case of this paragraph is possible to confront obtained results with other studies.

Response:  As mentioned above, we know of no other studies reported in which sows were fed formaldehyde-based feed sanitizers either in high health conditions or under a pathogen challenge so we have nothing to compare with.  No change made.

Lines 654-660, 693-700 and 707-712

The same as in lines 581-592.

Response:  See comment above for Line 581-592.

Lines 668-673

Lactobacillus sp. or spp?

Response:  Correction made.

Line 418

Genus: Peptoniphilus sp. or spp?

Response:  This is a repeat of the comment above.

Lines 716-723

Genus or Sarcina spp.

Response:  Correction made.

Line 725

Lactobacillus ruminis and Lactobacillus reuteri (italics required).

Response:  Correction made.

Lines 743-749

The same as in lines 581-592.

Response:  Same response as lines 581-592.

Lines 818-902

Abbreviation/s of Journal’s name must be used.

Response:  We apologize for the oversight.  Proper journal abbreviations have been included in the citations.  This applies to the following 3 comments.

Line 904

Sci. Rep.

Lines 909, 913 and 919

Abbreviation/s of Journal’s name must be used.

Line 911, 935, 942 and 947

Dot/point on the end of each abbreviation of Journal’s name required.

Lines 934-953

Information about purpose of references [A] – [I] can be added in header.

Response:  These citations are duplicates and have been removed.  Sorry for them slipping through on the submitted version.

Reviewer 2 Report

Comments and Suggestions for Authors

Dear authors,

thank you very much for submitting your manuscript. Please find the Reviewer's report down below.

Kind regards

Reviewer's Report

Summary:

In this study the authors examine the effects of the feed sanitizer Termin-8® on sow and piglet performance as well as gut microbiome during gestation and suckling period until weaning. They were able to show significant changes in the microbiome of sows over time from gestation to weaning. Effects on piglets’ microbiome, the performance of sows, the litter size, feed intake or the weight at weaning.

General comment on the hypothesis of the work:

The present study is very interesting and provides important information on the effects of the sanitizer Termin-8® on sows and piglets during gestation and suckling period.

The simple summary and the abstract are short and precisely written.

The introduction needs more detailed information on the effects of feed sanitation on pathogens and mycotoxins.

In the Materials and Methods section, further additions must be made to materials and methods used. See detailed comments down below.

In the results the formatting of the tables and figures should be improved.

The discussion is very detailed and well written. However, influences of the high ambient temperature (24 ± 1°C) of the farrowing stalls must be discussed. The thermoneutral temperature of sows varies between 18-19°C and heat stress starts at temperatures of 22°C. Especially heat stress has significant effects on gut health and systemic inflammation in pigs.

In the references are additional literature listed (A-I). Please explain these references.

Comments on the Abstract:

L18: piglets instead of babies

L48: gut microbiome (The term microbiome is too superficial. You analyze the gut microbiome.)

Comments on Introduction:

LL57-58: Please specify the most important mycotoxins in swine and briefly describe the effects on health of pigs. In addition describe effects of possible pathogens like Salmonella spp. and their burden in feed.

L66: Please add a reference.

LL71-73: bacterial families instead of bacteria species. (You listed bacterial families not species)

Comments on Material and Methods:

L93: Waseca, MN, USA.

L96: You wrote that sows of mixed parity were used. Please specify the overall span of parity of the sows.

LL100-101: You wrote that the sows were selected based on parity. Which parity parameters were used for the selection?

L174: Escherichia (E.). coli (italic style)

LL172-174: Why did you only screen for E. coli? Other pathogens like Clostridium perfringens or viral diseases could also causes diarrhea in piglets.

L182: Lawrenceville, GA, USA

L189: Please separate the subtitle and the paragraph.

LL193-194: Please add manufacturer, city, federal state and national state for the sterile collection tube.

L206: Please separate the subtitle and the paragraph.

LL208-210: Are there standardized antibiotic treatments in the other piglets? In this case, please add the treatments to the description of husbandry.

L226: Please add city, federal state and national state for the Spectrophotometer.

L246: Please add manufacturer, city, federal state and national state for the statistical program R Studio.

L249: Please separate the subtitle and the paragraph.

L250: Cary, NC, USA

L261: Please separate the subtitle and the paragraph.

LL339, 340: E. coli (italic style)

LL345, 348, 349: The deviation is very high. Are the numbers of the reads correct?

L348: Round bracket too much.

L353: Round bracket and dot too much.

Table 1:

The tryptophan content is very low. Conventional rations often contain 0.18% tryptophan. What is the total tryptophan content in the different rations?

Please add the abbreviations SID and CP to the description of the table and write out.

Please add the content of crude fiber in the table.

Comments on the Results:

LL318-319: Treatment sows

L319: post-weaning

LL329-330: Did you analyze the weight of mummies in relation to the litter size per sow?

L390: gestation

L391: increased

L393: an unknown (space too much); Lactobacillus (italic style)

L395: gestation

L399: Mollicutes (italic style)

L400: abundance; an unknown (missing space); Peptinophilus (italic style)

L401: ASV was (missing space); group

L409: q ≤ 0.005 (missing space)

L452: L. ruminis (Please explain the abbreviation)

L460: gestation

L463: difference

L466: Treatment sow

L491: compostion; treatment

L496: different

Table 3:

Please add the abbreviations SEM, Trt, Trt*Time to the description of the table and write it out.

Please add the national state to manufacturer of Termin-8®.

Table 4:

Please add the p-values for the “Weaning-to-estrus interval to 14d” and “21d”.

The description “Number of sows” is only marked for one trait.

Table 5:

If the table is continued on another page, please add the table header row and the information “Cont. Table 5”.

Please write a note in unlabeled cells. E.g. not applicable (NA) or not significant (n.s.).

Please add the national state to manufacturer of Termin-8®.

Please add the abbreviations SEM, Trt, Trt*Time to the description of the table and write it out.

Figure 1:

Please add the abbreviation ASV to the description.

Supplemental Table 2:

R² Treatment group

Please add the abbreviations F, R² and Pgroup to the description of the table and write it out.

 Comments on the discussion:

L668: Lactobacillus spp.; Lactobacillales spp. (italic style)

L673: Lactobacillus spp. (italic style)

L676: Lactobacillus spp. Instead of Lactobacilli

L680: Lactobacillales spp.

L715: Sarcina spp.

LL725-726: Lactobacillus ruminis (italic style); L. reuteri (italic style, Please explain the abbreviation)

L740: Lactobacillus (italic style)

Other detailed comments:

L793: Treatment diet

L809: Lawrenceville, GA, USA; St. Paul, MN, USA

L812: “ is too much

L819: Anim Feed Sci Technol

L821: J Agron

L832: Crit Rev Toxicol

L837: J Anim Sci

L846: Appl Eng Agric

L857: ISME J

L870: Environ Res

L875: Atmos Res

L887: Anim Reprod Sci

L899: Anim Nutr

LL901-902: Transl Anim Sci

L909: J Transl Intern Med

L913: J Nutr

L919: Food Funct

Author Response

Summary:

In this study the authors examine the effects of the feed sanitizer Termin-8® on sow and piglet performance as well as gut microbiome during gestation and suckling period until weaning. They were able to show significant changes in the microbiome of sows over time from gestation to weaning. Effects on piglets’ microbiome, the performance of sows, the litter size, feed intake or the weight at weaning.

General comment on the hypothesis of the work:

The present study is very interesting and provides important information on the effects of the sanitizer Termin-8® on sows and piglets during gestation and suckling period. The simple summary and the abstract are short and precisely written.

The introduction needs more detailed information on the effects of feed sanitation on pathogens and mycotoxins.

Response:  We have added additional text in the introduction (L58-64) to address this suggestion. 

In the Materials and Methods section, further additions must be made to materials and methods used. See detailed comments down below.

In the results the formatting of the tables and figures should be improved.

The discussion is very detailed and well written. However, influences of the high ambient temperature (24 ± 1°C) of the farrowing stalls must be discussed. The thermoneutral temperature of sows varies between 18-19°C and heat stress starts at temperatures of 22°C. Especially heat stress has significant effects on gut health and systemic inflammation in pigs.

Response:  The room temperature reported is incorrect.  There was an error in converting room temperature in degrees F to degrees C.  Since we think in degrees F, the error did not jump out at us as it did to this reviewer.  Our apologies.  We have corrected the temperature to 20 degrees C in the manuscript. 

In the references are additional literature listed (A-I). Please explain these references.

Response:  These references are an artifact of earlier versions of the manuscript.  All these references are included in the list above.  References A-I have been deleted.

Comments on the Abstract:

L18: piglets instead of babies

Response:  Suggested change made.

L48: gut microbiome (The term microbiome is too superficial. You analyze the gut microbiome.)

Response:  Suggested change made.

Comments on Introduction:

LL57-58: Please specify the most important mycotoxins in swine and briefly describe the effects on health of pigs. In addition describe effects of possible pathogens like Salmonella spp. and their burden in feed.

Response:  We have added some text here in response to Reviewer 1’s request.  However, this reviewer is asking for quite a bit of information that is tangentially related to the focus of this paper.  Adding this information in enough detail to be meaningful will greatly lengthen the introduction and not substantiatively contribute to the story we are trying to convey to the reader.  Consequently, we did not include additional text above that requested by Reviewer 1. 

L66: Please add a reference.

Response:  This statement is our interpretation of the text earlier in this paragraph so there is no citation for it.

LL71-73: bacterial families instead of bacteria species. (You listed bacterial families not species)

Response:  Suggested changes made.

Comments on Material and Methods:

L93: Waseca, MN, USA.

Response:  Suggested change made.

L96: You wrote that sows of mixed parity were used. Please specify the overall span of parity of the sows.

Response:  Range of parities has been added.

LL100-101: You wrote that the sows were selected based on parity. Which parity parameters were used for the selection?

Response:  We used parity number (0-8) in the assignment of sows to dietary treatments.  We attempted to equalize parity across dietary treatments as much as possible. 

L174: Escherichia (E.). coli (italic style)

Response:  Suggested change made.

LL172-174: Why did you only screen for E. coli? Other pathogens like Clostridium perfringens or viral diseases could also causes diarrhea in piglets.

Response:  Good point.  You are correct.  A full characterization of pathogens that might be responsible for diarrhea in piglets was beyond the scope of this project.  In retrospect, the extremely low incidence of diarrhea in piglets in this study would not have yielded much additional information if we analyzed for a wider range of pathogens.

L182: Lawrenceville, GA, USA

Response:  Suggested change made

L189: Please separate the subtitle and the paragraph.

Response:  Suggested change made

LL193-194: Please add manufacturer, city, federal state and national state for the sterile collection tube.

Response:  Requested information has been included.

L206: Please separate the subtitle and the paragraph.

Response:  Suggested change made

LL208-210: Are there standardized antibiotic treatments in the other piglets? In this case, please add the treatments to the description of husbandry.

Response:  There was not a standardized antibiotic treatment of all piglets.

L226: Please add city, federal state and national state for the Spectrophotometer.

Response:  Suggested change made

L246: Please add manufacturer, city, federal state and national state for the statistical program R Studio.

Response:  R is free software downloadable from the internet.  There is no traditional manufacturer or corporate headquarters.  Consequently, we included the website in the text where one can access the software

L249: Please separate the subtitle and the paragraph.

Response:  Suggested change made

L250: Cary, NC, USA

Response:  Suggested change made

L261: Please separate the subtitle and the paragraph.

Response:  Suggested change made

LL339, 340: E. coli (italic style)

Response:  Suggested change made

LL345, 348, 349: The deviation is very high. Are the numbers of the reads correct?

Response:  Yes, note that samples that yielded fewer than 1,000 reads were eliminated. The final reads range (range: 1,295 to 108,934) is not uncommon for Ilumina MiSeq projects, given inherent biological variations in microbial biomass among samples

L348: Round bracket too much.

Response:  Extra bracket was removed.

L353: Round bracket and dot too much.

Response:  Extra text removed

Table 1:

The tryptophan content is very low. Conventional rations often contain 0.18% tryptophan. What is the total tryptophan content in the different rations?

Response:  We disagree.  According to NRC (2012), the SID Trp requirement for gestating sows ranges from 0.07 to 0.13 for parity 1 through 4 sows at various levels of productivity.  For lactating sows, the range in SID Trp requirements is 0.13 to 0.17.  The calculated SID Trp contents of 0.10 and 0.16 for gestating and lactating sows, respectively is well within these guidelines. 

Please add the abbreviations SID and CP to the description of the table and write out.

Response:  CP is now spelled out in the table.  SID and STTD are defined in table footnotes.

Please add the content of crude fiber in the table.

Response:  Crude fiber for control gestation and lactation diets has been added to Table 1. 

Comments on the Results:

LL318-319: Treatment sows

Response:  We do not understand what the reviewer is asking for here.  No change made.

L319: post-weaning

Response:  Suggested change made.

LL329-330: Did you analyze the weight of mummies in relation to the litter size per sow?

Response:  We did not.

L390: gestation

Response:  Correction made

L391: increased

Response:  Correction made

L393: an unknown (space too much); Lactobacillus (italic style)

Response:  Correction made

L395: gestation

Response:  Correction made

L399: Mollicutes (italic style)

Response:  Suggested change made

L400: abundance; an unknown (missing space); Peptinophilus (italic style)

Response:  Suggested changes made

L401: ASV was (missing space); group

Response:  Suggested changes made

L409: q ≤ 0.005 (missing space)

Response:  Suggested changes made

L452: L. ruminis (Please explain the abbreviation)

Response:  Lactobacillus has been written out.

L460: gestation

Response:  Correction made

L463: difference

Response:  Correction made

L466: Treatment sow

Response:  We are not clear on what the reviewer is asking here

L491: compostion; treatment

Response:  Corrections made

L496: different

Response:  Correction made

Table 3:

Please add the abbreviations SEM, Trt, Trt*Time to the description of the table and write it out.

Please add the national state to manufacturer of Termin-8®.

Response:  Suggested changes made

Table 4:

Please add the p-values for the “Weaning-to-estrus interval to 14d” and “21d”.

The description “Number of sows” is only marked for one trait.

Response:  The p-value displayed is for the entire analysis.  This is a categorical analysis of dietary treatment by post-weaning period.  This analysis determines if the distribution of estrus was random or affected by dietary treatment at 7, 14, or 21 days after weaning.  The Chi Square analysis reports only one p-value.  It seems intuitive that the numbers in the body of the table are counts of sows hence there is no need to put the “2” superscript on every table entry.  No changes made.

Table 5:

If the table is continued on another page, please add the table header row and the information “Cont. Table 5”.

Response:  We are happy to do this but that only makes sense if we know exactly the final layout of the paper.  We assume there were be some changes (large or small) to the layout in the final typesetting process that might break the table at a different point than the current version.  We assume the Technical Editor will advise us on this.

Please write a note in unlabeled cells. E.g. not applicable (NA) or not significant (n.s.).

Response:  Again, we are happy to comply with this request but it seems that the table will be unduly cluttered with a tremendous amount of useless information.  For instance, in the repeated measures analysis of litter weight, there are no p-values for diet, time, and the interaction for each timepoint when litters were weighed.  There is an overall p-value for these effects which we report.  We are happy to take guidance from the Technical Editor on this issue.

Please add the national state to manufacturer of Termin-8®.

Response:  Done

Please add the abbreviations SEM, Trt, Trt*Time to the description of the table and write it out.

Response:  Done

Figure 1:

Please add the abbreviation ASV to the description.

Response:  Done

Supplemental Table 2:

R² Treatment group

Please add the abbreviations F, R² and Pgroup to the description of the table and write it out.

Response:  We think this is not necessary for F and R2 since these are common statistical terms and we assume they are in the journal’s list of standard abbreviations (although we could not find that list).  Pgroup is currently defined in the footnote to Table S2.

Comments on the discussion:

L668: Lactobacillus spp.; Lactobacillales spp. (italic style)

L673: Lactobacillus spp. (italic style)

L676: Lactobacillus spp. Instead of Lactobacilli

L680: Lactobacillales spp.

L715: Sarcina spp.

LL725-726: Lactobacillus ruminis (italic style); L. reuteri (italic style, Please explain the abbreviation)

L740: Lactobacillus (italic style)

Response:  These corrections (L668-740) have been made throughout the manuscript

Other detailed comments:

L793: Treatment diet

Response:  Correction made

L809: Lawrenceville, GA, USA; St. Paul, MN, USA

Response:  Suggested change made

L812: “ is too much

Response:  Suggested change made

L819: Anim Feed Sci Technol

L821: J Agron

L832: Crit Rev Toxicol

L837: J Anim Sci

L846: Appl Eng Agric

L857: ISME J

L870: Environ Res

L875: Atmos Res

L887: Anim Reprod Sci

L899: Anim Nutr

LL901-902: Transl Anim Sci

L909: J Transl Intern Med

L913: J Nutr

L919: Food Funct

Response:  Journal abbreviations have been updated to an acceptable format.

Reviewer 3 Report

Comments and Suggestions for Authors

Dear Authors,

This is a great and important research that you have conducted. Please find below the small corrections that I suggest to be made:

Line 18 - In my opinion, it's better to write sows and piglets instead of mothers and their babies. I think it's more professional.

Line 21 - Explain whether piglets are traditionally weaned on the 19th day after birth in your country?

Line 25 - After words of stillborn piglets you can add which positively affects production efficiency.

Line 48 - Instead of word swine I will put word piglet, since you already stated in the title that it's about sows and piglet.

Line 115 - How many sows were there per pen?

Line 133 - Was anesthesia used during castration, and what disinfectant did you use?

Line 168 - Instead of word scours, maybe it is better to write diarrhea. It is only stated as advice.

Line 329 - which also affects the economics of pig production.

Line 530 - Was the concentration of feed sanitizer tested only through chemical analysis of food or feces in sows and piglets? In your opinion, could the presence of feed sanitizer in muscle tissue, for example, be determined by sacrificing several piglets? The question is related to the aspect of the safety of such meat for human consumption.

Best regards

Author Response

This is a great and important research that you have conducted. Please find below the small corrections that I suggest to be made:

Line 18 - In my opinion, it's better to write sows and piglets instead of mothers and their babies. I think it's more professional.

Response:  We agree.  Change made.

Line 21 - Explain whether piglets are traditionally weaned on the 19th day after birth in your country?

Response:  Typical weaning age of piglets in the US ranges from 18 to 23 days of age.  Our weaning age is very common in the US.

Line 25 - After words of stillborn piglets you can add which positively affects production efficiency.

Response:  We do not see the direct connection between reduced weight of stillborn pigs and improved production efficiency.  There could be improved efficiency if one assumes that the reduced weight represents less loss of nutrients invested during gestation and therefore a greater proportion of the nutrient investment by the pregnant sow in piglets is realized at birth.  However, this is a very subtle, nuanced interpretation that does not have space to be explained in the abstract.  Hence, we decided to not include this phrase in the abstract.

Line 48 - Instead of word swine I will put word piglet, since you already stated in the title that it's about sows and piglet.

Response:  Suggested change made.

Line 115 - How many sows were there per pen?

Response:  Phrase was added to the text to indicate that about 55 sows were housed in each pen.

Line 133 - Was anesthesia used during castration, and what disinfectant did you use?

Response:  As per standard practice in the U.S., no anesthesia was used at castration.  The disinfectant used was iodine.

Line 168 - Instead of word scours, maybe it is better to write diarrhea. It is only stated as advice.

Response:  Thank you for the suggestion. 

Line 329 - which also affects the economics of pig production.

Response:  We agree.  However, stating this idea in this location seems to be a thought without context and would disrupt the flow of the paper. 

Line 530 - Was the concentration of feed sanitizer tested only through chemical analysis of food or feces in sows and piglets?  In your opinion, could the presence of feed sanitizer in muscle tissue, for example, be determined by sacrificing several piglets? The question is related to the aspect of the safety of such meat for human consumption.

Response:  The concentration of sanitizer was only measured in feed through chemical analysis.  We did not measure any tissues in sows or piglets to determine formaldehyde concentration. 

Response:  In my opinion, the likelihood of tissue residues is very, very small given the rapid metabolism of formaldehyde in the body as we describe in L59-66.

Round 2

Reviewer 1 Report

Comments and Suggestions for Authors

Thank you for the response for comments and suggestions.

I don't have any more. 

Reviewer 2 Report

Comments and Suggestions for Authors

Dear authors,

thank you very much for incorporating my comments and answering my questions. I have no further comments and the paper can be published in the present form.

Kind regards.